# CHART-R1: CHAIN-OF-THOUGHT SUPERVISION AND REINFORCEMENT FOR ADVANCED CHART REASONER

## ABSTRACT

Recently, inspired by OpenAI-o1/o3 and Deepseek-R1, the R1-style method based on reinforcement fine-tuning has received widespread attention from the community. Previous R1-style methods mainly focus on mathematical reasoning and code intelligence. It is of great research significance to verify their advantages on more general multimodal data. Chart is an important multimodal data type with rich information, which brings important research challenges in complex reasoning. In this work, we introduce Chart-R1, a chart-domain vision-language model with reinforcement fine-tuning to enable complex chart reasoning. To support Chart-R1, we first propose a novel programmatic data synthesis technology to generate high-quality step-by-step chart reasoning data covering single- and multi-subcharts, which makes up for the lack of reasoning data in the chart domain. Then we develop a two-stage training strategy: Chart-COT with step-by-step chain-of-thought supervision, and Chart-RFT with numerically sensitive reinforcement fine-tuning. Chart-COT aims to decompose complex chart reasoning tasks into fine-grained, understandable subtasks through step-by-step supervision, which lays a good foundation for improving the reasoning capacity of reinforcement learning. Chart-RFT utilizes the typical group relative policy optimization strategy, in which a relatively soft reward is adopted for numerical response to emphasize the numerical sensitivity in the chart domain. We conduct extensive experiments on open-source benchmarks and a self-built chart reasoning dataset (*i.e., ChartRQA*). Experimental results show that Chart-R1 has significant advantages compared to chart-domain methods, even comparable to open/closed source large-scale models (*e.g., GPT-4o, Claude-3.5*).

## 1 INTRODUCTION

Recently, inspired by the success of models such as OpenAI's o1/o3 OpenAI (2025b) and DeepSeek-R1 Guo et al. (2025), leveraging Reinforcement Learning (RL) for fine-tuning has garnered significant attention within the research community. Although these methods have shown promise in textual domains like mathematical reasoning, code generation, and multidisciplinary knowledge, transferring these advanced reasoning capabilities to the vision domain presents an open challenge. While recent approaches like Vision-R1 Huang et al. (2025) and VLM-R1 Shen et al. (2025) have successfully leveraged RL to enhance visual perception and grounding, they have primarily focused on simple questions, neglecting tasks that demand deep reasoning capabilities.

Charts, as information-intensive images, are a crucial research area in image understanding and reasoning Wang et al. (2024). Prior works improve chart perception and understanding capacities by supervised fine-tuning (SFT) on datasets augmented with Chain-of-Thought (CoT) or Program-of-Thought (PoT) methods Wei et al. (2022); Chen et al. (2022). A key limitation of SFT is that it causes models to overfit specific reasoning patterns, hindering their generalization abilities. Following the DeepSeek R1, recent methods Jia et al. (2025); Ni et al. (2025) leverage RL to enhance VLM reasoning capabilities. However, the scope of these efforts has been largely limited to visual perception and understanding, rather than the complex reasoning required for deep chart analysis.

In this work, we propose Chart-R1, a chart domain VLM that leverages RL to enhance complex reasoning capability, which achieves superior performance as shown in Figure 1. To this end, we introduce two key contributions. First, we propose a novel programmatic synthesis strategy to generate

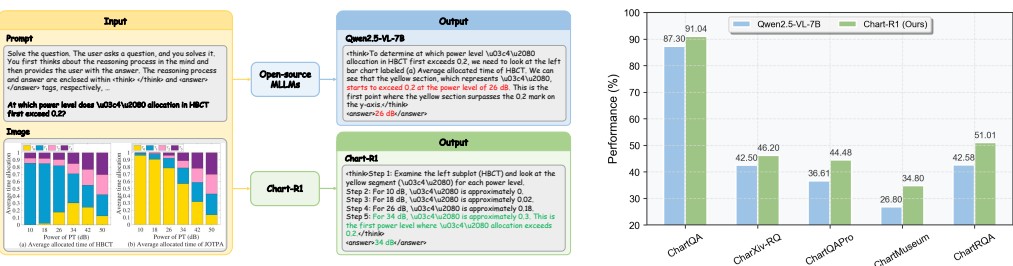

Figure 1: Comparison of Qwen2.5-VL-7B and Chart-R1 on chart understanding and reasoning benchmarks. In the complex chart reasoning task, Qwen2.5-VL-7B generates a wrong thinking process, whereas Chart-R1 thinks and answers correctly.

high-quality reasoning data. Second, we introduce an effective two-stage training strategy that significantly enhances reasoning capacity. Specifically, to support Chart-R1 training, we first generate complex chart reasoning data in the programmatic synthesis method. We utilize LLMs to generate the chart plotting code and then use the generated code to formulate complex questions, multi-step CoT reasoning processes, and the final answer. To this end, we construct ChartRQA, a complex reasoning dataset featuring 258k multi-step reasoning samples that cover both single- and multi-chart tasks. To ensure the fidelity of the data in charts, we curate real-world tables from arXiv papers as the data source. The training of Chart-R1 is conducted in two stages: Chart-COT with step-by-step chain-of-thought supervision, and Chart-RFT with numerically sensitive reinforcement fine-tuning. During the initial Chart-COT stage, the model is fine-tuned via SFT on step-by-step reasoning data to build its core capability of decomposing complex tasks into fine-grained subtasks. In the Chart-RFT stage, we use the group relative policy optimization (GRPO) strategy with a composite reward signal of soft matching and edit distance to enhance accuracy for both numerical and string-based answers. We employ distinct datasets for these two stages, as our findings show that training on the same data impairs the model's exploration ability during the RL process. Furthermore, we introduce ChartRQA, a human-verified benchmark designed to probe the limits of complex chart reasoning. In contrast to prior works Wang et al. (2024), its questions demand a higher degree of complexity and multi-step thought processes. The substantial performance drop of existing VLMs on ChartRQA reveals a critical gap in their reasoning capabilities. In summary, our contributions are as follows:

- To enhance chart reasoning in VLMs, we propose a novel two-stage training strategy consisting of Chart-COT and Chart-RFT. The resulting model, Chart-R1, sets a new state-of-the-art across various chart understanding and reasoning benchmarks.

- We introduce a programmatic data synthesis strategy that leverages code as a pivotal starting source to generate step-by-step reasoning data. The data source is grounded in real-world tables from arXiv papers, ensuring data fidelity in the resulting charts.

- We introduce ChartRQA, a comprehensive dataset for complex chart reasoning that includes a human-verified benchmark and a large-scale training dataset. The substantial performance drop of existing VLMs on the ChartRQA benchmark underscores a critical limitation in their chart reasoning capabilities.

## 2 RELATED WORKS

### 2.1 CHART VLMS

Chart understanding and reasoning are crucial areas of research community that encompass both low-level and high-level tasks Singh et al. (2019); Methani et al. (2020). Recently, many chart-domain models have been proposed to enhance the chart understanding capacity of VLMs Han et al. (2023); Liu et al. (2023). However, prior works have concentrated on descriptive tasks Masry et al. (2024a;b), such as extracting explicit content from charts Masry et al. (2022). In contrast, more recent works focus on leveraging the reasoning capabilities of VLMs to interpret complex and implicit information within the charts. For example, TinyChart Zhang et al. (2024) utilizes a template-based method to generate the Program-of-Thought (PoT) Chen et al. (2022) reasoning data. ChartCoder Zhao et al. (2025b) proposes Snippet-of-Thought to enhance chart-to-code generation. ChartReasoner Jia et al. (2025) utilizes a chart-to-code model to convert chart images into code and

generate the reasoning process based on code. However, the generated reasoning data has limitations due to the chart-to-code accuracy Shi et al. (2024); Xu et al. (2024).

## 2.2 Long Reasoning VLMs

Recently, with the success of DeepSeek-R1 Guo et al. (2025), many works have attempted to enhance the LLM reasoning ability via rule-based reward and RL Shao et al. (2024). In the vision-language domain, recent works follow the DeepSeek-R1 method to enhance the long-chain reasoning capacity of VLMs Shen et al. (2025); Wang et al. (2025); Qiu et al. (2025). For example, Vision-R1 Huang et al. (2025) and R1-OneVision Yang et al. (2025) apply Group Relative Policy Optimization (GRPO) with multimodal reasoning data to enable VLMs for long reasoning. MMEureka Meng et al. (2025b) and R1-Zero Liu et al. (2025) further advance the visual long-term reasoning with improved RL training strategies. Point-RFT Ni et al. (2025) uses grounded CoT for visual understanding, but it just utilize ChartQA for RL which limits the final model reasoning capacity.

## 2.3 Chart Understanding and Reasoning

A variety of training datasets and evaluation benchmarks have been developed to improve VLM performance on chart-related tasks Xia et al. (2024); Shi et al. (2024); He et al. (2024); Zhao et al. (2025a); Wu et al. (2025). Previous works generally focus on description tasks, for example, ChartQA Masry et al. (2022), PlotQA Methani et al. (2020) and Chart-to-text Kantharaj et al. (2022) mainly train and evaluate the capacities of the models on extracting information from the chart. While numerous relevant works exist, the challenge in the description tasks is predominantly driven by chart complexity. Recent works such as Charxiv Wang et al. (2024) and CharMuseum Tang et al. (2025a) introduce more challenging reasoning tasks, demanding that models think before answering. Unlike descriptive tasks, reasoning tasks present a dual challenge, originating from both the perceptual complexity of the chart and the reasoning depth required by questions.

## 3 Method

To enhance the reasoning capabilities of models on chart reasoning tasks, we introduce our proposed data synthesis and two-stage training strategy. We first programmatically generate a large-scale training dataset with the CoT reasoning process and subsequently employ the SFT on CoT data as a cold start phase to bootstrap the subsequent RL strategy for training.

## 3.1 Programatic Data Synthesis

While several CoT datasets for chart reasoning have been proposed, they are largely derivatives of the ChartQA dataset, constructed by augmenting its existing question-answer pairs with generated reasoning processes Zhang et al. (2024); Jia et al. (2025). However, this method is like distilling reasoning from top VLMs, so it naturally inherits their limitations and errors on complex tasks. The reliance on final answer correctness as the only supervisory signal makes generating high-quality CoT reasoning data a significant challenge. This issue is amplified in complex chart reasoning, where the struggles of even top models inherently lead to low-quality, undiverse data. Although the recent ChartReasoner method Jia et al. (2025) generates reasoning data by first parsing charts into code, the diversity and quality of generated data are fundamentally limited by the performance of the chart-to-code parser. In contrast, our programmatic data generation strategy reverses this paradigm by utilizing code as a pivotal starting source. First, we prompt a powerful LLM to generate plotting code. This code then serves as a perfect, high-fidelity foundation from which a VLM subsequently synthesizes question-answer pairs and their complex step-by-step reasoning path. An overview of our data synthesis pipeline is shown in Figure 2.

**Plotting Code Generation** We instruct LLMs to generate Matplotlib plotting code to render high-quality and diverse chart images. However, our analysis reveals that directly generating synthetic data values in plotting code often yields monotonous trends that lack complexity and diversity. To address this, we first curate tables from real-world arXiv papers, which serve as veritable data sources. Secondly, to enhance the diversity of the generated code, we manually write seed code examples for different chart types. To ensure the diversity of generated code, we randomly combine

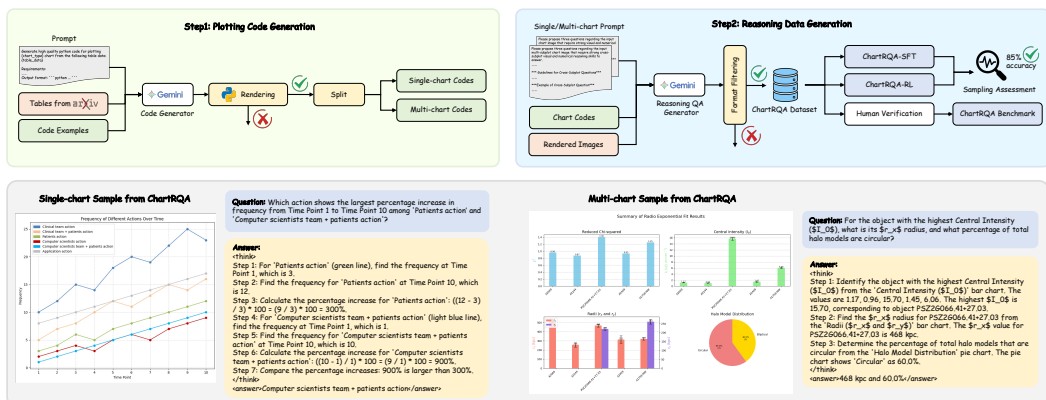

Figure 2: ChartRQA dataset pipeline via programmatic data synthesis. The pipeline consists of two main stages: plotting code generation and reasoning data generation, resulting in ChartRQA-SFT, ChartRQA-RL, and the ChartRQA benchmark. The bottom part of the figure presents examples from both single-chart and multi-chart scenarios in ChartRQA, featuring complex questions that require step-by-step thinking processes to answer.

the curated table and seed code as in-context learning sources for LLMs to generate plotting code. To generate complex, multi-chart scenarios, we both include numerous multi-chart examples in our seed code and explicitly prompt the LLM during generation to use functions like plt.subplots() to create composite figures. Our work significantly expands the range of chart types available for chart reasoning, representing the most diverse dataset. We execute all generated code samples and discard any that fail to run successfully.

**Reasoning Data Generation** With the executable plotting code as a foundation, we prompt LLMs to synthesize a complete reasoning instance, comprising a question, its answer, and a step-by-step reasoning path. To enhance diversity, we categorize the plotting code into single- and multi-chart types and apply distinct generation instructions for each. For multi-chart problems, we instruct the LLM to generate questions that necessitate cross-referencing information between sub-charts. The generated data show that this strategy significantly enhances multi-chart task complexity. Our results show that leveraging code allows LLMs to produce more complex questions and detailed reasoning compared to methods that use chart images alone. We posit that a code-based approach is superior for generating complex chart reasoning as the underlying code provides a lossless textual representation while enabling the scalable synthesis of data independent of existing corpora.

**Dataset Construction** Using the aforementioned methods, we construct the ChartRQA dataset, which includes a large-scale training dataset of 258k instances with reasoning paths as well as a human-verified benchmark. The training dataset is separated into two subsets for our two-stage training strategy, ChartRQA-SFT and ChartRQA-RL, consisting of 228k and 30k samples, respectively. The benchmark is constructed via a human validation where experts review each sample for question difficulty and answer correctness, subsequently constructing 1,702 high-quality samples (933 single-chart and 769 multi-chart tasks) for evaluation. As detailed in Table 1, we also calculated the average token counts for the questions, reasoning paths, and final answers, broken down by single- and multi-chart problems. The analysis reveals that the components associated with multi-chart problems are significantly longer than those for single-chart problems.

**Quality Evaluation** To assess the quality of our generated data, we randomly sample 1k instances and recruit human experts for evaluation. The results indicate that over 85% of the instances are free from errors. Notably, we deliberately omit any data cleaning process. The fact that our model, Chart-R1, achieves strong performance despite being trained on this raw, uncurated dataset validates the robustness of our proposed code-based generation strategy.

### 3.2 CHART-COT

To enhance the chart reasoning capacity, we propose a two-stage training strategy. Utilizing Qwen2.5-VL-7B-Instruct as the baseline model, we first SFT it on the step-by-step reasoning data of our proposed ChartRQA-SFT. Specifically, the baseline model first undergoes SFT on our gen-

Table 1: The average question, thinking process, and answer lengths in the ChartRQA train and test sets. We count the single- and multi-chart problems of each set separately.

| Token Avg. | Train | | | Test | | |
|---|---|---|---|---|---|---|
| | Single | Multi | Total | Single | Multi | Total |
| Question | 30.03 | 39.84 | 34.03 | 29.83 | 39.49 | 34.19 |
| Thinking Process | 196.50 | 237.38 | 213.17 | 196.32 | 240.94 | 216.48 |
| Answer | 5.98 | 8.87 | 7.16 | 5.96 | 8.97 | 7.32 |

erated step-by-step reasoning data, which serves as the code-starting phase to equip the model with the fundamental capability to decompose complex tasks into fine-grained subtasks. Our ablation studies demonstrate that a preliminary SFT stage on CoT data is critical, as it yields significantly better performance than applying RL from scratch.

We train the model using a standard autoregressive language modelling objective. The loss function is the negative log-likelihood of the target sequence:

$$\mathcal{L}(\theta) := -\mathbb{E}_{(x,y)\sim\mathcal{D}_{\text{CoT}}} \sum_{t=1}^{T} \log P\left(y_t \mid x, y_{<t}; \theta\right), \tag{1}$$

where $(x, y)$ is the query and target response, with the reasoning process.

### 3.3 CHART-RFT

After the Chart-COT stage, while the fine-tuned model demonstrates an enhanced ability to decompose complex questions, its performance on out-of-domain (OOD) tasks notably degrades. We hypothesize this is due to a distributional mismatch between ChartRQA-SFT with some simple chart understanding tasks, which harms its generalization ability. To address the degradation in generalization, we subsequently apply reinforcement fine-tuning (RFT) to generalize its reasoning capacity.

**Group Relative Policy Optimization** We adapt the Group Relative Policy Optimization (GRPO) algorithm Shao et al. (2024); Guo et al. (2025), which significantly conserves training resources by replacing the critic model with a baseline estimated from group scores. For each input $(x, y)$, the policy $\pi_\theta$ samples a group of $G$ candidate responses $\{o_i\}_{i=1}^{G}$.

$$\mathcal{J}_{GRPO}(\theta) = \mathbb{E}_{\{o_i\}_{i=1}^{G}\sim\pi_{\theta_{\text{old}}}(O|x)} \left[ \frac{1}{G} \sum_{i=1}^{G} \min\left( \frac{\pi_\theta(o_i \mid x)}{\pi_{\theta_{\text{old}}}(o_i \mid x)} A_i, \right. \right. \tag{2}$$

$$\left. \left. \text{clip}\left( \frac{\pi_\theta(o_i \mid x)}{\pi_{\theta_{\text{old}}}(o_i \mid x)}, 1 - \varepsilon, 1 + \varepsilon \right) A_i \right) \right]$$

where $\varepsilon$ is the hyperparameter, $\pi_\theta$ and $\pi_{\theta_{\text{old}}}$ are the optimized model and the policy model respectively. The group-normalized advantage for the $i$-th response is:

$$A_i = \frac{r_i - \text{mean}\left(\{r_1, r_2, \cdots, r_G\}\right)}{\text{std}\left(\{r_1, r_2, \cdots, r_G\}\right)} \tag{3}$$

**Reward Design** For effective RFT, we follow the DeepSeek-R1 Shao et al. (2024) and adopt a rule-based reward that consists of accuracy and format rewards. The reward function consists of two parts: the accuracy and format rewards for assessing the answer and format correctness, respectively.

- **Accuracy Reward.** We employ distinct, type-specific reward functions to measure the correctness of model outputs. For numerical answers, we adopt the soft matching technique from Point-RFT Ni et al. (2025) with a relative error tolerance of $\pm 5\%$. For string-based answers, we utilize the edit distance as the reward signal.

- **Format Reward.** The format reward is determined by a grammar-level regex parser that validates the structural integrity of outputs. It confirms two conditions: (1) the reasoning process is properly enclosed in <think> tags, and (2) the final answer is extractable from the designated <answer> tags.

**Data Proportion** For the Chart-COT and Chart-RFT stages, we utilize distinct subsets of ChartRQA. This setting is critical, as our experiments reveal that using the same CoT data for

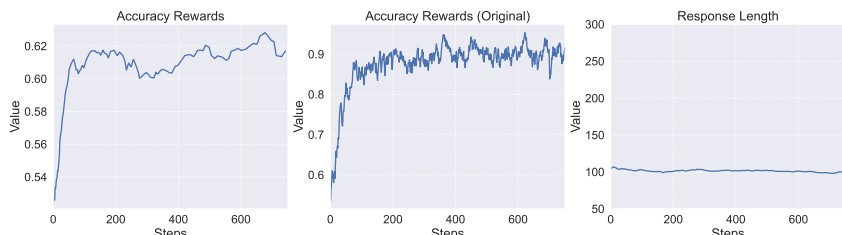

Figure 3: The training curve during the RL stage that utilizes the ChartQA dataset solely.

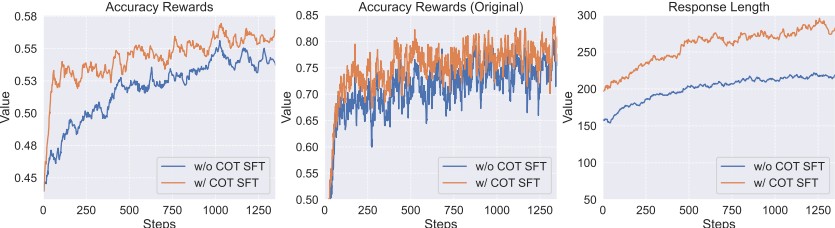

Figure 4: Training curves for the RL stage using the ChartQA and ChartRQA datasets. The orange curve represents our proposed two-stage training strategy, while the blue curve corresponds to a baseline RL-only setting.

both phases causes the model to overfit to replicate the reasoning paths from the SFT data, which in turn degrades the diversity and exploration capability of the policy model during the RL phase. We find that the stability and convergence of the Chart-RFT phase critically depend on the pattern consistency of the data from the preceding Chart-COT stage. Employing SFT data with inconsistent patterns significantly hinders RFT convergence, highlighting the necessity of a distributionally aligned dataset in the Chart-COT stage to ensure effective downstream RFT.

## 4 EXPERIMENTS

### 4.1 EXPERIMENT SETTINGS

We conduct experiments and ablation studies to evaluate the results obtained from various training settings. See Appendix A.1 for training details.

**Benchmarks** To comprehensively evaluate the understanding and reasoning capacity of our posed Chart-R1, we choose ChartQA Masry et al. (2022), CharXiv-RQ (Reasoning Questions) Wang et al. (2024), ChartQAPro Masry et al. (2025a), ChartMuseum Tang et al. (2025a) and our proposed ChartRQA (single/multi) as the evaluation benchmarks.

**Baselines** We compare our proposed Chart-R1 with existing models in three setups: (1) Proprietary models include GPT-4o, GPT-4.1 OpenAI (2025a), Gemini-1.5-(Flash, Pro), Gemini-2.5-Flash Kavukcuoglu (2025), Claude-3.5-Sonnet and Claude-3.7-Sonnet Anthropic (2025). (2) General-domain open-source VLMs including Phi 3.5-Vision Abdin et al. (2024), DeepSeek-VL2 Wu et al. (2024), InternVL3(8B, 38B) Zhu et al. (2025) and Qwen2.5-VL(7B) Bai et al. (2025). (3) Chart-domain VLMs including ChartLlama Han et al. (2023), TinyChart Zhang et al. (2024), Chart-Gemma Masry et al. (2024b), ChartResoner Jia et al. (2025), BigCharts-R1 Masry et al. (2025b) and Bespoke-MiniChart-7B Tang et al. (2025b).

### 4.2 MAIN RESULTS

Table 2 shows the performance of Chart-R1 compared with other baseline models. The results show that Chart-R1 achieves the state-of-the-art performance on small-scale (<20B) VLMs, including general- and chart-domain models across various chart understanding and reasoning benchmarks. Especially in ChartQA, Chart-R1 achieves the best performance, even compared with proprietary and large-scale VLMs. In the chart reasoning benchmark, CharXiv-RQ, ChartMuseum and our proposed ChartRQA, Chart-R1 significantly surpass existing chart-domain models. Since the training data of Chart-R1 only contains ChartRQA and ChartQA, these results demonstrate the effectiveness of our proposed ChartRQA dataset and CoT-RL training strategy.

Table 2: The main results on existing chart understanding and reasoning benchmarks. Our proposed Chart-R1 achieves superior performance among small-scale VLMs ($<$20B) on the evaluation benchmarks. **Bold** denotes the best performances of open-source VLMs.

| Model Name | ChartQA | CharXiv-RQ | ChartQAPro | ChartMuseum | ChartRQA (single / multi) |
|---|---|---|---|---|---|
| *Proprietary* | | | | | |
| GPT-4o | 85.7 | 47.1 | 37.67 | 42.2 | 44.37 / 46.55 |
| Gemini-1.5-Flash | 79.0 | 33.9 | 42.96 | 31.1 | - |
| Gemini-1.5-Pro | 87.2 | 43.3 | - | 41.3 | - |
| Gemini-2.5-Flash | - | - | - | - | 59.12 / 59.17 |
| Claude-3.5-Sonnet | 90.8 | 60.2 | 43.58 | 54.4 | 52.79 / 56.05 |
| GPT-4.1 | 86.8 | 56.7 | - | 48.4 | 57.88 / 59.30 |
| Claude-3.7-Sonnet | 86.1 | 64.2 | - | 60.3 | 55.04 / 57.87 |
| *General-domain Open-source* | | | | | |
| Phi-3.5-Vision | 81.8 | 32.7 | 24.73 | - | 31.08 / 24.32 |
| DeepSeek-VL2 | 86.0 | - | 16.28 | - | 23.15 / 20.29 |
| InternVL3-8B | 86.6 | 37.6 | - | 28.2 | 37.51 / 31.73 |
| InternVL3-38B | 89.2 | 46.4 | - | 32.1 | 46.09 / 38.36 |
| Qwen2.5-VL-7B | 87.3 | 42.5 | 36.61 | 26.8 | 44.59 / 40.57 |
| *Chart-domain* | | | | | |
| ChartLlama | 69.66 | 14.2 | - | - | - |
| TinyChart | 83.60 | 8.3 | 13.25 | 12.5 | 6.75 / 6.11 |
| ChartGemma | 80.16 | 12.5 | 6.84 | 12.2 | 7.18 / 9.23 |
| ChartReasoner | 86.93 | - | 39.97 | - | - |
| BigCharts-R1 | 89.84 | 41.3 | - | - | - |
| Bespoke-MiniChart | 89.50 | 45.4 | **45.36** | 34.0 | 42.77 / 42.13 |
| Chart-R1 (Ours) | **91.04** | **46.2** | 44.48 | **34.8** | **52.09 / 49.93** |

Table 3: The ablation study about different SFT and RL training settings. QA and RQA are the abbreviations of ChartQA and ChartRQA.

| Model Name | Training Setting | | ChartQA | CharXiv-RQ | ChartRQA (single / multi) |
|---|---|---|---|---|---|
| | SFT | RL | | | |
| Qwen2.5-VL-7B | | | 87.3 | 42.5 | 44.59 / 40.57 |
| Qwen2.5-VL-7B-SFT | *QA* | | 86.16 | 36.0 | 24.76 / 18.34 |
| Qwen2.5-VL-7B-RL | | *QA* | 89.32 | 42.1 | 37.73 / 36.15 |
| | | *QA+RQA-RL* | 90.28 | 45.2 | 44.16 / 40.44 |
| | *RQA-SFT* | *QA+RQA-RL* | **91.04** | **46.2** | **52.09 / 49.93** |

## 4.3 ABLATION STUDY

We first assess the impact of different training settings, with results presented in Table 3. The findings indicate that utilizing our two-stage training strategy yields the most balanced performance. Notably, omitting Chart-COT causes a significant performance drop on the ChartRQA benchmark. We attribute this to complex charts requiring multi-step thinking before answering. The first Chart-COT stage equips the model with the necessary capability for such step-by-step task decomposition. Also, SFT on the ChartQA dataset leads to performance degradation across all benchmarks, including ChartQA itself. We reckon that although SFT could improve capacity for in-domain tasks, training on simple and low-diversity datasets disrupts the tuned distribution, harming the ability on both in-domain (ChartQA) and OOD (CharXiv-RQ, ChartRQA) tasks.

Prior research underscores the critical role of training data complexity for effective RL Guo et al. (2025). Our generated ChartRQA training set addresses this by featuring tasks with both single- and multi-chart images, and questions demanding step-by-step reasoning. Including our ChartRQA dataset during the RL stage is crucial for achieving optimal performance. The structural and logical complexity is important for performance enhancements observed in our Chart-RFT stage. Furthermore, RL exclusively on the ChartQA dataset is insufficient for developing a reasoning model. The

Table 4: More ablation studies within RL and SFT stages. ED and SM are the abbreviations of Edit Distance and Soft Matching. RQA[†] indicates that only samples with chart types of line, bar, and pie in ChartRQA are used for training.

(a) Ablation study on accuracy reward, and training set in RL.

| RL Setting | ChartQA | CharXiv-RQ |
|---|---|---|
| *Accuracy Reward* | | |
| ED | 89.88 | 44.0 |
| ED + SM | **90.28** | **45.2** |
| *RL Training Set* | | |
| QA | 89.32 | 42.1 |
| QA + RQA[†] | **90.32** | 44.6 |
| QA + RQA | 90.28 | **45.2** |

(b) Ablation study on training set and different datasets (TinyChart and ChartGemma) in SFT.

| SFT Setting | ChartQA | CharXiv-RQ |
|---|---|---|
| *SFT Training Set* | | |
| RQA-SFT&RL + QA | 88.40 | 41.2 |
| RQA-SFT | **89.88** | **44.5** |
| *SFT Dataset* | | |
| TinyChart | 84.80 | 36.1 |
| ChartGemma | 86.72 | 39.1 |
| ChartRQA-SFT | **90.20** | **45.0** |

limited complexity of ChartQA fails to encourage the model to learn diverse, long-path reasoning strategies. This limitation is empirically demonstrated by the training process shown in Figure 3. The accuracy reward rapidly converges to around 0.9 with little subsequent growth, while the response length remains constrained to approximately 100 tokens.

We further investigate the impact of our two-stage training strategy, comparing it to a baseline without the Chart-COT phase. The comparison of RL processes is shown in Figure 4. We find that the first SFT on CoT data has two key benefits. First, it significantly increases the token length generated during the RL phase. Second, it leads to a much effective accuracy reward curve, which rises quickly at the start of training and then converges at a higher final value.

**Reward Function** To assess different accuracy rewards, we conduct experiments by training Qwen2.5-VL-7B-Instruct for the RL stage only, as shown in Table 4a. The results demonstrate that employing a soft accuracy reward, which combines edit distance for string-based tasks and soft matching for numerical tasks, yields superior performance across both benchmarks. This finding underscores the importance of adjusting the reward function to the specific type of answers.

**Image Diversity & Question Complexity** Our ChartRQA dataset is characterized by two key features: diverse chart images and complex questions requiring step-by-step reasoning. To investigate the importance of these factors in RL training, we select samples from ChartRQA-RL that only include line, bar, and pie chart types, which are the same chart types found in the ChartQA dataset, and train the model using RL only. As shown in Table 4a, without increasing chart type diversity, the complex questions in ChartRQA still substantially enhance the model's reasoning ability. Furthermore, using the full ChartRQA dataset, which includes a wider variety of chart images, leads to further improvements on CharXiv-RQ.

**SFT Data Composition** When training Chart-R1, our SFT dataset consists of 228k samples from our ChartRQA-SFT. We then ablate the SFT data composition by adding two sources, the ChartQA dataset and the 30k ChartRQA-RL that overlaps with the RL data, to assess the impact on performance. We train each setting for 2k steps and 1 epoch for SFT and RL, respectively. The results in Table 4b show that combining ChartQA and ChartRQA-RL, the final performance decreases evidently. Our analysis indicates that using overlapping data for SFT and RL leads to overfitting, where the model memorizes reasoning paths from the SFT stage, resulting in more rigid thinking processes and a significant loss of output diversity. Also, the direct-answer format of ChartQA discourages the model from developing the ability to break down problems into a step-by-step thinking process.

**Comparison with Existing Chart Datasets** To enable a fair comparison between ChartRQA and existing chart SFT datasets, we replace the SFT dataset with TinyChart and ChartGemma, while keeping all other settings consistent. TinyChart is a comprehensive dataset that integrates multiple open-source datasets and comprises a variety of tasks. To ensure that the model focuses on chart understanding and reasoning, we exclude the Chart-to-text and Chart-to-table generation tasks. For the RL stage, we use a combination of ChartQA and ChartRQA-RL for 1 epoch of training. As shown in Table 4b, Chart-R1 trained on ChartRQA-SFT achieves the best performance on both benchmarks. The results indicate that the unified thinking and answer format and the effective step-by-step reasoning process in ChartRQA are key factors in enhancing the model's reasoning ability.

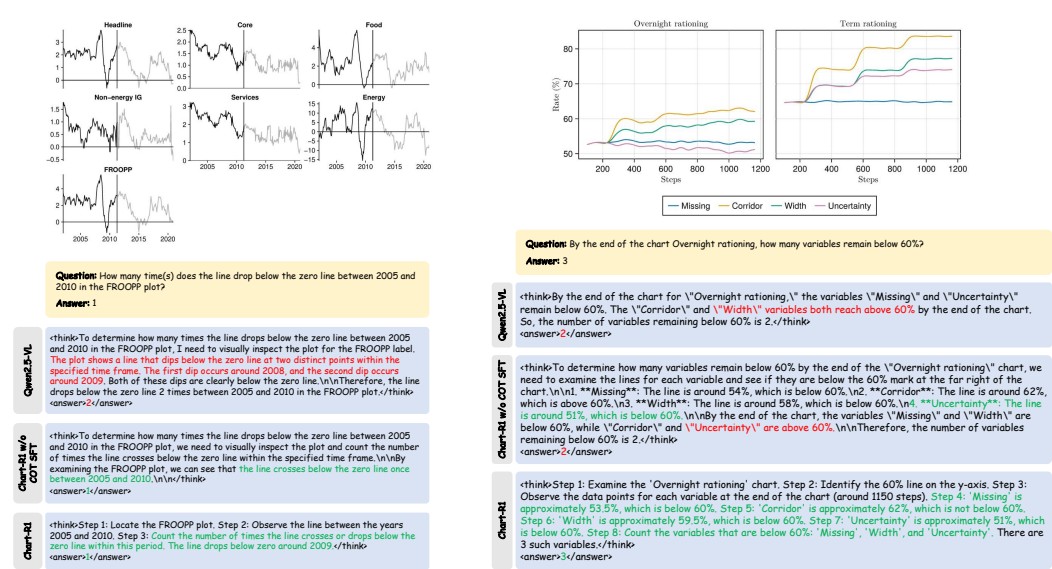

Figure 5: Visualization results of two case studies: (a) Chart-R1 (with and without Chart-COT) both answer correctly while Qwen2.5-VL-7B fails, and (b) only Chart-R1 with Chart-COT answers correctly while both Qwen2.5-VL-7B and Chart-R1 without Chart-COT fail.

**Visualization** We present qualitative case studies where our Chart-R1 model successfully generates detailed reasoning and correct answers for complex questions in Figure 5. In these same instances, the baseline Qwen2.5-VL-7B model fails, directly demonstrating the superior performance and more advanced reasoning capabilities of our approach. When Chart-R1 is trained without the Chart-COT stage, it also fails to answer the problems in the right case of Figure 5. Although it can correctly recognize the chart content, it makes errors during the reasoning process, highlighting the importance of our proposed two-stage training.

## 4.4 ERROR ANALYSIS

Chart-R1 achieves significant improvements in reasoning ability compared to baseline, but there is still room for further improvement. We randomly sample 50 incorrect responses from Chart-R1 on ChartQAPro and analyze the error types. As shown in Figure 6, Chart-R1 is most prone to errors in visual reasoning, multi-chart QA, and unanswerable types. Visual reasoning is more challenging than mathematical reasoning, as the latter mainly involves numerical recognition and calculation, while the former requires the model to identify and summarize complex chart patterns. Multi-chart QA requires the model to integrate information across multiple charts. While ChartRQA was designed to address multi-chart reasoning, the current model still exhibits deficiencies in this aspect. For unanswerable questions, although ChartRQA did not specifically include such samples, Chart-R1 can reject most unanswerable questions through reinforcement learning, demonstrating good generalizability.

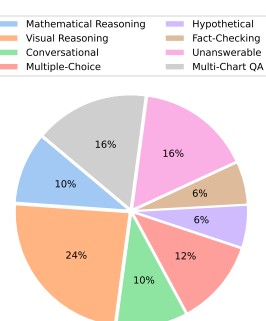

Figure 6: Error type distribution of Chart-R1 on ChartQAPro.

## 5 CONCLUSION

In this paper, we propose Chart-R1, a chart-domain VLM for complex chart reasoning. To improve the reasoning capacity of Chart-R1, we introduce a programmatic data generation method alongside a novel two-stage training strategy to optimize the data construction and training methodology. Also, we propose ChartRQA, which contains 258k training samples, each constructed in verifiable formats and a benchmark for evaluating complex chart reasoning. The result shows that combining our proposed training strategy, Chart-R1 achieves superior performance compared with other VLMs.

## 6 REPRODUCIBILITY STATEMENT

For datasets, we provide a detailed description of our data generation process in Section 3.1, complemented by dataset statistics and examples in Appendix A.3. The exact prompts used for dataset construction are available in Appendix A.4. For implementation, we include Chart-R1's complete SFT and RL training code, along with model inference examples, in the supplementary materials.

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

# A APPENDIX

## A.1 TRAINING DETAILS

In this section, we provide the implementation details of Chart-R1's training process, including Chart-COT and Chart-RFT.

**Chart-COT** We use Qwen2.5-VL-7B-Instruct as the initial model and perform supervised fine-tuning using LLaMA-Factory Zheng et al. (2024). We train the model on the 228k ChartRQA-SFT dataset for one epoch. During training, we freeze the vision tower and multi-modal projector parameters and tune the LLM. The learning rate is set to 1e-5, with a warm-up ratio of 0.1 and batch size of 48. The training process costs 3 hours on 24 H800 GPUs.

**Chart-RFT** For the RFT stage, we use the fine-tuned model from the Chart-COT stage. We adopt the MM-EUREKA Meng et al. (2025a) framework based on OpenRLHF for training. The model is trained for 3 episodes using 30k ChartQA and 30k ChartRQA-RL. We set the rollout batch size and the training batch size to 128, with each sample generating 8 rollouts. The temperature for model generation is set to 1, and we exclude KL divergence in the loss calculation. The learning rate is set to 1e-6, with a warm-up ratio of 0.03, while freezing the vision tower during training. Following the default setting for instruction models, the format reward coefficient is set to 0.5. We employ the online filtering strategy with lower and upper bounds of 0.1 and 0.9, respectively. The training process costs 30 hours on 24 H800 GPUs.

## A.2 BENCHMARK DETAILS

**ChartQA** Masry et al. (2022) focuses on chart question answering with complex reasoning questions that involve logical and arithmetic operations. Following the settings in the original paper, we evaluate models on the test set reporting overall accuracy scores across both human-written (ChartQA-H) and machine-generated (ChartQA-M) question subsets.

**CharXiv** Wang et al. (2024) presents a comprehensive evaluation suite with natural, challenging, and diverse charts from arXiv papers to provide a more realistic assessment of chart understanding capabilities. We evaluate models on the Reasoning Questions (CharXiv-RQ) subset, which requires synthesizing information across complex visual elements in charts. Following the original paper, we use GPT-assisted evaluation to assess model responses.

**ChartQAPro** Masry et al. (2025a) introduces a diverse benchmark with various chart types, including infographics and dashboards, and question formats that better reflect real-world challenges. We evaluate models using Chain-of-Thought (CoT) prompting in the original paper and report overall accuracy across five question types.

**ChartMuseum** Tang et al. (2025a) is a chart question-answering benchmark designed to evaluate complex visual reasoning capabilities with expert-annotated questions from diverse real-world charts. Following the original paper, we evaluate models using the provided CoT prompt and LLM-as-a-Judge evaluation method.

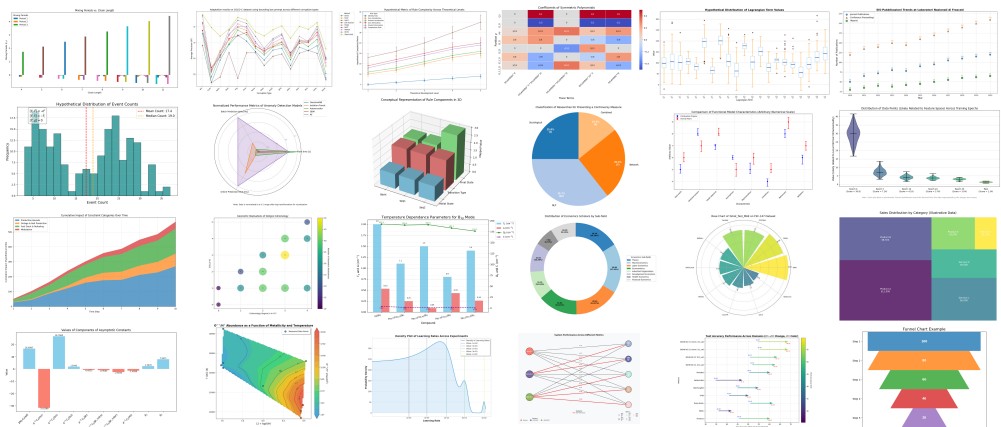

Figure A: Single-chart samples of 24 chart types from ChartRQA.

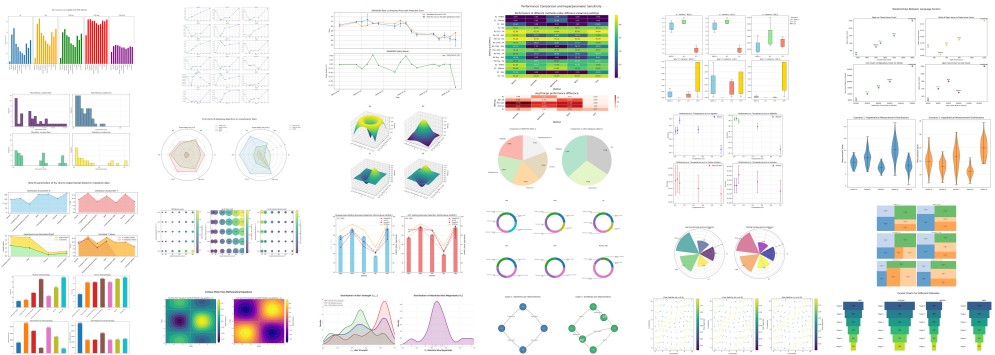

Figure B: Multi-chart samples of 24 chart types from ChartRQA.

## A.3 CHARTRQA ANALYSIS

Table A presents a detailed comparison between ChartRQA and other existing chart-domain datasets. ChartRQA stands out by integrating a broader range of chart types, supporting multi-chart reasoning, and providing step-by-step thinking annotations, which collectively contribute to its effectiveness for advanced chart reasoning tasks. We further analyze the quantity and distribution of different chart types across the training and test sets of ChartRQA, as detailed in Table B. The distribution among the various types to be well-balanced. Furthermore, Figures A and B provide visualization examples of 24 chart types from the ChartRQA dataset, showcasing both single-chart and multi-chart formats, respectively.

## A.4 PROMPTS

To enhance transparency and reproducibility, we provide the exact prompts used for dataset generation and evaluation. For data generation, we employ Gemini-2.5-Flash to generate both plotting

Table A: Comparison of our proposed ChartRQA training set with other chart datasets. ChartRQA features the integration of single/multi-charts, thinking processes, and verifiable answer formats.

| Dataset | Types | Unique Charts | Multi-chart | Thinking Process |
|---|---|---|---|---|
| ChartQA Masry et al. (2022) | 3 | 21.9k | ✗ | ✗ |
| MMC Liu et al. (2023) | 7 | 600k | ✔ | ✗ |
| ChartLlama Han et al. (2023) | 10 | 11k | ✗ | ✗ |
| NovaChart Hu et al. (2024) | 18 | 47k | ✗ | ✔ |
| ChartRQA (Ours) | 24 | 93.3k | ✔ | ✔ |

Table B: The detailed chart types and corresponding quantities in our proposed ChartRQA train and test set. ChartRQA contains 24 chart types, each of which contains approximate samples.

| Split | Bar | Line | ErrorBar | Heatmap | Box | Scatter | Histogram |
|---|---|---|---|---|---|---|---|
| Train | 11,850 | 10,752 | 11,838 | 8,993 | 12,112 | 10,299 | 15,856 |
| Test | 100 | 88 | 83 | 60 | 103 | 76 | 116 |

| Split | Radar | 3D | Pie | ErrorPoint | Violin | Area | Bubble |
|---|---|---|---|---|---|---|---|
| Train | 9,483 | 9,746 | 17,812 | 10,814 | 12,571 | 9,175 | 8,996 |
| Test | 46 | 65 | 103 | 68 | 116 | 75 | 51 |

| Split | Multi-axes | Ring | Rose | Treemap | Bar_num | Contour | Density |
|---|---|---|---|---|---|---|---|
| Train | 10,776 | 12,726 | 10,533 | 9,850 | 12,150 | 10,291 | 12,860 |
| Test | 61 | 54 | 61 | 64 | 64 | 67 | 77 |

| Split | Graph | Quiver | Funnel | Total | | | |
|---|---|---|---|---|---|---|---|
| Train | 8,764 | 9,955 | 227 | 258,429 | | | |
| Test | 47 | 52 | 5 | 1,702 | | | |

code and QA pairs for data construction. Figure C illustrates the prompt used for plotting code generation. We utilize real table data as input, select one chart type from the 24 predefined chart types, and sample a code example corresponding to that chart type to generate the plotting code. Figures D and E display the prompts used to generate reasoning QA pairs for single-chart and multi-chart formats, respectively. We craft an example for each format to aid LLMs in understanding complex chart reasoning tasks and to generate step-by-step reasoning processes and precise answers that conform to the format. The executable plotting code is provided as auxiliary information to LLMs, making the generated QA pairs more reliable. Figure F shows the prompt used for model evaluation. We employ GPT-4o to assess the match between the ground truth and the model's predictions, where GPT-4o returns a score of 0 or 1 to indicate the correctness of the model's prediction. Our evaluation focuses solely on the correctness of the final answer, disregarding the reasoning process.

## A.5 THE USE OF LARGE LANGUAGE MODELS (LLMS)

In this work, we used LLMs as writing tools to improve language clarity and readability. These models helped refine the text and enhance the presentation of our ideas. All research concepts, experiments, implementations, and analyses were conducted independently by the authors.

## A.6 LIMITATIONS AND FUTURE WORK

Our study focuses primarily on statistical charts from academic papers, overlooking practical visualization types such as dashboards and flowcharts. This leads to a gap compared to closed-source models on comprehensive benchmarks such as ChartMuseum. In future research, we plan to expand our training paradigm to incorporate diverse chart types and complex visual reasoning, developing a more versatile chart understanding model.

---

**Prompt for Code Generation**

Generate high quality python code for plotting {chart_type} chart from the following table data:
{table_data}

Requirements:
The code must present table data in a reasonable way.
The code example of {chart_type} chart (given in JSON format) is:
{code_example}

You must not be limited by the code sample and draw different styles of dials.
The generated code should not be too complicated and all text elements (labels, titles, legends) must be fully visible without overlap or truncation.
Pie/Ring/Treemap chart visualization: always display the actual numerical values on each segment. Percentages are optional, but values must be clearly visible.
IMPORTANT: Generate only ONE figure with all necessary information. If multiple plots are needed, use subplots (plt.subplots) to arrange them in a single figure.
Output format: ```python ... ```

Figure C: Prompt for code generation.

---

**Prompt for Reasoning QA Pairs Generation (Single-chart)**

Please propose three questions regarding the input chart image that require strong visual and numerical reasoning skills to answer. These questions should involve multi-step reasoning processes that challenge analytical abilities. Provide detailed answers with step-by-step reasoning. The reasoning process and final answer should be enclosed within <think> and <answer> tags, respectively.

Below is the Python code used to generate this chart. You can use this as reference, but your questions and answers should be based on the visual elements and data actually displayed in the chart image:
```python
{python_code}
```

***Guidelines for Effective Reasoning Questions***
1. Questions should require 2-5 reasoning steps to solve
2. Include questions about relationships between different data points or series
3. Ask about mathematical operations (differences, percentages, ratios) between data elements
4. Focus on identifying patterns, extremes, or anomalies in the data visualization

***Example of a Strong Reasoning Question***
Question: What is the sum of the max value of Series A and the min value of Series B?
Answer:
<think>
Step 1: First, identify all values of Series A in the chart. The values are [23, 45, 32, 18, 50].
Step 2: The maximum value of Series A is 50.
Step 3: Next, identify all values of Series B in the chart. The values are [42, 38, 45, 40, 41].
Step 4: The minimum value of Series B is 38.
Step 5: Finally, calculate the sum: 50 + 38 = 88.
</think>
<answer>
88
</answer>

Please strictly adhere to the information displayed in the image when posing questions and providing answers. The answers should be obtainable solely through observation of the image. Avoid posing open-ended questions, and ensure a definite answer using a single word or phrase for each question. Do not fabricate questions or propose questions requiring external knowledge to solve.

Your response should strictly follow the format below and be returned in JSON format:
[{{"Question": "Your first question here...", "Answer": "<think>Your first thinking process here...</think><answer>Your first answer here...</answer>"}}, {{"Question": "Your second question here...", "Answer": "<think>Your second thinking process here...</think><answer>Your second answer here...</answer>"}}, {{"Question": "Your third question here...", "Answer": "<think>Your third thinking process here...</think><answer>Your third answer here...</answer>"}}]

Figure D: Prompt for reasoning QA pairs generation for single-chart formats.

---

**Prompt for Reasoning QA Pairs Generation (Multi-chart)**

Please propose three questions regarding the input multi-subplot chart image that require strong cross-subplot visual and numerical reasoning skills to answer. These questions must necessitate analyzing and integrating information from multiple subplots to arrive at the correct answer. Provide detailed answers with step-by-step reasoning processes. The reasoning process and final answer should be enclosed within <think> and <answer> tags, respectively.

Below is the Python code used to generate this multi-subplot chart. You can use this as reference, but your questions and answers should be based on the visual elements and data actually displayed across all subplots in the chart image:
```python
{python_code}
```

***Guidelines for Cross-Subplot Questions***
1. Each question MUST require information from at least two different subplots to answer correctly
2. Questions should involve comparisons, relationships, or integrations across different subplots
3. Include questions that require mathematical operations (e.g., differences, ratios, correlations) between data from multiple subplots
4. Focus on identifying patterns, trends, or anomalies that are only visible when considering multiple subplots together

***Example of Cross-Subplot Question***
Question: If we compare the maximum value in subplot A with the average value in subplot B, what is their percentage difference?
Answer:
<think>
Step 1: Identify the maximum value in subplot A. Looking at the first subplot, I can see that the maximum value is 85.
Step 2: Calculate the average value in subplot B. In the second subplot, the values are [42, 38, 45, 40, 41], so the average is (42+38+45+40+41)/5 = 206/5 = 41.2.
Step 3: Calculate the percentage difference: ((85-41.2)/41.2)*100 = (43.8/41.2)*100 = 106.31%
</think>
<answer>
106.31%
</answer>

Please strictly adhere to the information displayed across all subplots when posing questions and providing answers. The answers should be obtainable solely through observation of the image. Avoid posing open-ended questions, and ensure a definite answer using a single word or phrase for each question. Do not fabricate questions or propose questions requiring external knowledge to solve.

Your response should strictly follow the format below and be returned in JSON format:
[{{"Question": "Your first question here...", "Answer": "<think>Your first thinking process here...</think><answer>Your first answer here...</answer>"}}, {{"Question": "Your second question here...", "Answer": "<think>Your second thinking process here...</think><answer>Your second answer here...</answer>"}}, {{"Question": "Your third question here...", "Answer": "<think>Your third thinking process here...</think><answer>Your third answer here...</answer>"}}]

Figure E: Prompt for reasoning QA pairs generation for multi-chart formats.

---

**Prompt for ChartRQA Model Evaluation**

You will be given a question, a ground truth answer, and a model response. Your task is to compare the model response with the ground truth answer and assign a binary score (0 or 1). Please provide only the score without any explanations or additional text. If there is no model response provided, assign a score of 0.

Please follow these scoring rules:

### Scoring Rules ###

1. **For Terminology and Concepts:**
* Score 1: The model response and ground truth refer to the same concept or term, even if expressed differently (e.g., a and alpha; $R^2_{t,h,v,m}$ and R^2_t,h,v,m). Different ordering of terms is acceptable when multiple terms are requested.
* Score 0: Any term in the response differs meaningfully from the ground truth (e.g., ACC+ vs ACC; P-101 vs P=101).

Example 1.1:
* Question: What is the name of the curve that intersects y=\lambda exactly three times?
* Ground Truth: P56962
* Response: There is only one curve that intersects y=\lambda exactly three times. The name of the curve is P55762.
Score: 0

Example 1.2:
* Question: What is the letter of the subplot where all bars are above 35?
* Ground Truth: (b)
* Response: The letter of the subplot where all bars are above 35 is b.
Score: 1

2. **For Numerical Values:**
* Score 1: The numerical values in the response and ground truth are mathematically equivalent, even if expressed in different notations (e.g., 0.01 and 10^-2; 1500 and 1.5e3).
* Score 0: The numerical values differ in their actual value, regardless of notation.

Example 2.1:
* Question: What is the value of the red curve at t=10?
* Ground Truth: 0.01
* Response: The value of the red curve at t=10 is 0.012.
Score: 0

Example 2.2:
* Question: What is the value of the blue curve at t=50?
* Ground Truth: 1500
* Response: The value of the blue curve at t=50 is 1.5e3.
Score: 1

3. **For Descriptive Trends and Patterns:**
* Score 1: The response conveys the same semantic meaning as the ground truth (e.g., "increasing then decreasing" and "moving up then down"; "converge" and "move closer together").
* Score 0: The response conveys a different semantic meaning from the ground truth (e.g., "increasing then decreasing" vs "remain constant"; "converge" vs "diverge").

Example 3.1:
* Question: What is the trend of the red curve between t=10 and t=25?
* Ground Truth: increasing then decreasing
* Response: The red curve is increasing between t=10 and t=25.
Score: 0

4. **For Multiple-Choice or Predefined Options:**
* Score 1: The selected option in the response matches the ground truth exactly.
* Score 0: The selected option differs from the ground truth.

Example 4.1:
* Question: What interval among [0, 50], [50, 100], [100, 150], and [150, 200] contains the maximum value of the blue curve?
* Ground Truth: [50, 100]
* Response: The interval where the blue curve achieves the maximum value is [50, 100].
Score: 1

### Your Task ###
* Question: <|question|>
* Ground Truth: <|ground_truth|>
* Response: <|response|>

Score:

---

Figure F: Prompt for ChartRQA evaluation using GPT-4o.