# OpenReview forum: "Chart-R1: Chain-of-Thought Supervision and Reinforcement for Advanced Chart Reasoner"
_ICLR.cc/2026/Conference — ICLR 2026 Conference Withdrawn Submission_

### Official Review · Reviewer_ScnT · 2025-10-29

**Soundness:** 2
**Presentation:** 3
**Contribution:** 2
**Rating:** 4
**Confidence:** 3

**Summary:**

This paper proposes ChartRQA, a large-scale dataset for complex chart reasoning in the chart domain, and Chart-R1, a corresponding two-stage training framework. The ChartRQA dataset consists of a total of 258K training samples—228K for SFT (ChartRQA-SFT) and 30K for RL (ChartRQA-RL) along with a human-verified benchmark of 1,702 samples. To build this dataset, the authors employ a code-based generation pipeline where an LLM is instructed to produce chart plotting code inspired by real-world charts and tables from arXiv papers. The generated charts include both single- and multi-chart settings.For question–answer pair construction, the authors explicitly embed step-by-step reasoning paths to enhance the model’s reasoning capabilities, and multi-chart samples are designed to require cross-referencing between subplots. The proposed Chart-R1 framework involves a two-stage training strategy—an SFT stage using CoT (Chain-of-Thought) data, and an RL stage using GRPO (Group Relative Policy Optimization) with accuracy and format rewards.Using this framework with the Qwen2.5-VL-7B-Instruct model trained on ChartRQA and ChartQA datasets, the authors report performance gains over multiple baselines.

**Strengths:**

1. The task of complex chart reasoning is important and has significant practical value for users in real-world data analysis scenarios.
2. The open-source nature of the dataset can positively impact future research in the chart reasoning community.
3. The paper is clearly written and easy to follow.
4. The authors provide code and implementation details, ensuring reproducibility.
5. The dataset generation pipeline is carefully designed to cover diverse chart types and reasoning complexities using LLM-based code synthesis.

**Weaknesses:**

While the authors state that they incorporate human verification and real arXiv charts, the dataset construction still heavily relies on LLMs. Such reliance raises concerns that synthetic data may introduce a domain gap from real-world chart data and that LLM-inherent biases could influence the data distribution and question formulation. Furthermore, it is unclear whether the proposed dataset is overly tailored to the Chart-R1 framework, which could limit its generality when used with other reasoning methods. The second claimed contribution, Chart-R1, does not appear to be substantially novel compared to prior works. its advantage is not clearly justified based on Table 2 alone.

**Questions:**

1. When constructing ChartRQA, the authors rely on an LLM to generate chart plotting code using data from arXiv tables. Are the resulting charts and legends numerically and semantically realistic enough to reflect real-world distributions? How do the authors address the potential domain gap between synthetic and real-world charts?
2. Since the dataset heavily depends on LLM-generated charts and question–answer pairs, could LLM-inherent biases (e.g., over-representation of specific domains, repeated question patterns) affect data diversity?
3. To what extent does ChartRQA cover diverse domains and levels of reasoning depth? Are the chart topics limited to scientific data (e.g., from arXiv), or do they generalize to other domains such as economics or social sciences?
4. In Related Work (line 120), the authors reference Point-RFT but the methodological difference between Point-RFT and Chart-R1 is unclear. What distinguishes Chart-R1 from Point-RFT conceptually and technically? Is this difference substantial enough to be considered a main contribution?
5. Were all chart-domain baselines in Table 2 trained on the same datasets as Chart-R1 (ChartRQA + ChartQA)? For a fair comparison and to demonstrate the true effectiveness of Chart-R1, please report results where all methods are trained on identical datasets.
6. Is ChartRQA’s strong performance limited to the Chart-R1 framework? If ChartRQA is genuinely a strong reasoning dataset, other reasoning models should also benefit when trained on it.  To better assess the reasoning effectiveness of ChartRQA, please provide a comparison of the chart-domain methods in Table 2 trained solely on ChartQA versus those trained on ChartRQA, and quantify the improvement in reasoning performance.
7. Does ChartRQA require ChartQA to achieve strong results? Please provide an ablation comparing ChartRQA-only vs. ChartQA-only training for Chart-R1.
8. Table 4 presents various ablations but does not include results on the ChartRQA benchmark. Could the authors provide those numbers to validate whether improvements transfer to the main evaluation set?
9. In Related Work (line 109), the authors mention that previous works such as ChartReasoner, ChartMimic, and ChartMoE suffer from low chart-to-code generation accuracy. Does the LLM used for ChartRQA generation achieve higher code accuracy compared to those models? Why did those methods still choose to maintain an explicit chart-to-code module instead of directly generating code via LLMs?

---

### Official Review · Reviewer_qGkB · 2025-10-30

**Soundness:** 2
**Presentation:** 2
**Contribution:** 2
**Rating:** 4
**Confidence:** 4

**Summary:**

This paper presents Chart-R1, a multimodal large language model for deep chart reasoning. The model is trained in two stages: (1) Chart-COT, supervised fine-tuning with chain-of-thought data to teach structured reasoning; and (2) Chart-RFT, reinforcement tuning using GRPO with soft accuracy and format rewards to improve numerical precision and response consistency. A large synthetic dataset, ChartRQA (258k training and 1.7k verified test samples), is programmatically generated from executable plotting code to ensure realistic chart–data alignment. Experiments on ChartQA and CharXiv-RQ show state-of-the-art results among baseline models, with gains in chart understanding and reasoning tasks.

**Strengths:**

**High-Fidelity Dataset Generation via Code (ChartRQA).**  ChartRQA is generated from executable plotting code rather than static images, ensuring perfect alignment between data and visuals, greater diversity (24 chart types), and complex multi-chart reasoning tasks.

**Soft Accuracy Reward Design.** The reinforcement stage introduces *soft numeric* and *edit-distance* rewards, which better capture partial correctness and stabilize training, yielding superior performance across benchmarks.

**State-of-the-Art Open-Model Performance.** Chart-R1 achieves the best results among sub-20B open models and even outperforms some proprietary systems (e.g., GPT-4o, Claude 3.5) on key chart reasoning benchmarks like ChartQA and CharXiv-RQ.

**Weaknesses:**

**Limited base-model diversity.** Most experiments are on a single backbone (Qwen2.5-VL-7B). Broader validation across architectures/sizes would strengthen claims about generality.

**Incremental novelty of the training scheme.** The two-stage CoT + RL pipeline follows recent R1 style work. The main novelty appears to be dataset construction and chart-specific reward shaping, rather than a fundamentally new algorithm.

**Small-scale analyses.** Some ablations (e.g., reward components, stage interplay) are limited in scope. Deeper analyses would help isolate which choices most drive the gains.

**Questions:**

**About the reported ~85% data accuracy.** The paper states that roughly 85% of ChartRQA samples are error-free (per human checks). Could the authors comment on whether this quality level is adequate for reliable large-scale training? It would be helpful to discuss how the remaining ~15% noise might affect robustness and final performance, and whether any performance differences were observed when training on cleaner subsets.

**Code execution success rate.** The dataset retains only Matplotlib scripts that execute successfully. Could the authors report the overall execution success rate during synthesis (attempted vs. retained)? Providing this statistic would clarify the pipeline’s efficiency and scalability.

**Methodological novelty beyond data.** Given similarities to R1-style pipelines (e.g., DeepSeek-R1, BigCharts-R1), it would be valuable for the authors to delineate the key algorithmic or training innovations unique to Chart-R1 beyond dataset construction, for instance, which aspects of reward design, sampling, or curriculum differ materially, and the rationale for those choices.

**General-domain trade-offs.** Following extensive chart-centric training, how does Chart-R1 perform on broader multimodal reasoning outside the chart domain? Any quantitative evaluation or discussion of potential specialization–generalization trade-offs would be appreciated.

---

### Official Review · Reviewer_Xa84 · 2025-11-01

**Soundness:** 3
**Presentation:** 2
**Contribution:** 2
**Rating:** 2
**Confidence:** 4

**Summary:**

This paper proposes Chart-R1, a chart understanding and reasoning model built upon Qwen-2.5-VL-7B. The authors propose a two-stage training pipeline: supervised finetuning (SFT) followed by GRPO. During the SFT phase, the authors propose a technique called “programmatic data synthesis” that helps generate precise and well-grounded QA pairs (along with CoT). This is achieved by leveraging the original code of the chart image with teacher models like Gemini, rather than relying solely on the images themselves. For RL, the authors propose two reward functions: format and accuracy (with 5% tolerance). The resulting model, Chart-R1 achieves state-of-the-art results on multiple downstream benchmarks such as ChartQA, CharXiv, ChartQAPro, and ChartMuseum.

**Strengths:**

* The SFT dataset enhances Qwen2.5-vl performance across various chart understanding benchmarks, including ChartQA, Charxiv , and ChartQAPRO.


* The well-designed RL approach, incorporating appropriate reward functions, further elevates the model's performance, achieving state-of-the-art results on a diverse range of benchmarks.

**Weaknesses:**

* The data tables used to render the chart images are mainly sourced from arXiv which limits the diversity of the dataset (e.g., topics). Furthermore, generating QA pairs exclusively from the code, without incorporating images, may lead to a dataset with fewer visual questions and a greater emphasis on the data itself.

* In lines 142-156, the authors claim that their data synthesis method works better than existing approaches that either augments existing datasets with CoT or generates QA pairs from chart images/parsed tables. However, they did not conduct a fair experiment to prove their claims. Hence, I suggest two sets of experiments:
  * **Comparative Finetuning on Existing Datasets:** Finetune the same base model (qwen2.5 vl) on established open-source datasets (e.g., ChartGemma, TinyChart, ChartReasoner) and compare its performance to their SFT'd model.


  * **Comparative Finetuning on Synthesized Datasets:** Using identical seed data (tables or charts), create a small CoT QA dataset with their method and with existing approaches that utilize chart images or parsed tables. Then, finetune the same model (qwen2.5 vl) on these newly constructed datasets and compare the results.


* The proposed dataset generation approach “programmatic data synthesis” is not novel. Some previous papers such as ChartLlama, MMC, ChartInstruct, and ChartAssistant have already generated CoT style chart reasoning data using the underlying data/code rather than chart images. It would be helpful if the authors can elaborate a bit more on the novelty of their work compared to these.

**Questions:**

I think the citations are broken throughout the paper. I suggest the authors fix them to avoid confusion in the future. There are also a few types in different parts of the paper that need proofreading.

---

### Official Review · Reviewer_8XyV · 2025-11-03

**Soundness:** 3
**Presentation:** 4
**Contribution:** 3
**Rating:** 6
**Confidence:** 4

**Summary:**

This paper introduces Chart-R1, a chart-domain Vision-Language Model (VLM) that leverages Reinforcement Learning (RL) to significantly enhance its complex chart reasoning capability. The work addresses the gap where prior RL-based VLM methods (R1-style) neglected tasks requiring deep, multi-step reasoning, particularly in the information-intensive domain of charts. The main contributions are: 1) a novel programmatic data synthesis strategy that uses code as a pivotal source to generate high-quality, step-by-step Chain-of-Thought (CoT) data; 2) a complex, large-scale dataset, ChartRQA, which includes both training samples and a human-verified benchmark focused on single and multi-chart reasoning; and 3) a two-stage training strategy (Chart-COT, followed by Chart-RFT) that uses numerically sensitive reinforcement fine-tuning to achieve superior performance. The resulting model achieves new state-of-the-art results across various chart benchmarks.

**Strengths:**

1. **Novel and Robust Data Generation:** The programmatic data synthesis approach—generating the plotting code first and then using the code to formulate complex questions, reasoning paths, and answers—effectively overcomes the limitations of existing methods, which often rely on distilling reasoning from weaker models or are constrained by the accuracy of chart-to-code parsers. This methodology enables the creation of a high-quality, diverse dataset for complex reasoning.

2. **Effective Two-Stage Training:** The proposed Chart-COT and Chart-RFT two-stage strategy is well-justified. The initial SFT stage on CoT data (Chart-COT) successfully builds the model’s fundamental capacity to decompose complex tasks into fine-grained subtasks, which is demonstrated as critical for the stability and performance of the subsequent RL stage.

3. **Domain-Specific RL Refinement:** The utilization of a numerically sensitive reward signal within the Chart-RFT (GRPO) stage is a key technical strength. This specialized reward, combining soft matching and edit distance, is tailored to the precision required in the chart domain, enhancing both numerical and string-based answer accuracy.

4. **Creation of a Comprehensive Benchmark:** The introduction of ChartRQA, a new human-verified benchmark, is a significant contribution, explicitly targeting the complex, multi-step, and multi-chart reasoning tasks that challenge current VLMs.

**Weaknesses:**

1. More VLM Reasoning Models should be included: MMR1, VL-Cogito, VL-Rethinker, OpenVLThinker, R1-VL, and so on.

2. Missing Quantitative Comparison to Contemporary Reasoning VLMs: The paper successfully positions Chart-R1 as an advancement over general VLMs. However, the paper's results tables do not include a direct, quantitative head-to-head comparison of Chart-R1 against these contemporary VLM Reasoning models on the complex ChartRQA benchmark. Including these comparisons is necessary to definitively prove the necessity and efficiency of the Chart-R1 domain-specific approach.

**Questions:**

1. **Generalizability of Programmatic Data:** While the programmatic synthesis is excellent for quality and complexity, does the training process inherently introduce any bias towards reasoning patterns or data characteristics typical of the Matplotlib plotting code used? What is the model's zero-shot performance on charts generated from distinct sources (e.g., business reports, academic papers using different packages) that might exhibit different visual styles or data distributions?

Overall this is a solid work, and I don't have strong weakenss to say. I am willing to raise the final rating if all concerns are addressed.

---

### Official Review · Reviewer_JeLx · 2025-11-10

**Soundness:** 3
**Presentation:** 2
**Contribution:** 2
**Rating:** 4
**Confidence:** 3

**Summary:**

This paper introduces Chart-R1, a chart-domain vision-language model that integrates chain-of-thought (CoT) supervision with reinforcement fine-tuning (RFT) to enhance complex chart reasoning. Motivated by recent R1-style reinforcement methods, the authors aim to extend such approaches from mathematical reasoning to multimodal chart understanding. The framework consists of two key stages: (1) Chart-COT, which employs step-by-step CoT supervision to decompose chart reasoning into interpretable subtasks, and (2) Chart-RFT, which applies numerically sensitive reinforcement fine-tuning using a soft reward formulation for chart-based responses. To support training, the authors construct a programmatic data synthesis pipeline to generate diverse step-by-step chart reasoning data covering single- and multi-subchart settings. They also introduce ChartRQA, a benchmark and dataset designed for reinforcement training and evaluation in chart reasoning tasks. Extensive experiments demonstrate that Chart-R1 achieves state-of-the-art performance among small-scale (<20B) vision-language models on multiple benchmarks.

**Strengths:**

S1. The paper presents an adaptation of R1-style reinforcement learning to the chart reasoning. The overall implementation is coherent, and the two-stage structure (Chart-COT + Chart-RFT) is technically sound.

S2. Experiments cover multiple public and in-domain benchmarks, with consistent improvements over open-source and chart-specific baselines.

S3. The data synthesis pipeline and ChartRQA benchmark could be useful for future work, provided they are released.

**Weaknesses:**

W1. The main motivation is that R1-style reinforcement learning has not yet been applied to chart reasoning, which is not a strong research question in itself.

W2. The method closely follows existing R1-style frameworks with only minor domain-specific adjustments.

W3. While the paper repeatedly emphasizes complex chart reasoning, it never clearly defines what constitutes “complex.” The distinction between complex reasoning and simpler chart understanding tasks remains ambiguous.

**Questions:**

Q1. Please provide an operational definition of what you mean by complex chart reasoning. In your benchmarks, which types of questions are categorized as “complex reasoning” versus “simple understanding”? Additionally, how does your model’s performance differ across these two categories?

Q2. In your error analysis, you note that visual reasoning accounts for the largest proportion of errors. Could you provide concrete examples to illustrate these cases?

---

### Note · Authors · 2025-11-24

I have read and agree with the venue's withdrawal policy on behalf of myself and my co-authors.